# The Tsallis generalized entropy enhances the interpretation of transcriptomics datasets

**Nicolas Dérian[1,2], Hang-Phuong Pham[3], Djamel Nehar-Belaid[1,2,4], Nicolas Tchitchek[1,2], David Klatzmann[1,2], Vicaut Eric[5], Adrien Six[1,2]***

**1** Sorbonne Université, INSERM, UMR-S 959, Immunology-Immunopathology- Immunotherapy (i3), Paris, France, **2** AP-HP, Hôpital Pitié-Salpêtrière, Biotherapy (CIC-BTi) and Inflammation-Immunopathology-Biotherapy Department (i2B), Paris, France, **3** IL-TOO Pharma, Paris, France, **4** The Jackson Laboratory for Genomic Medicine, Farmington, CT, United States of America, **5** APHP, Hôpitaux Saint-Louis Lariboisière, Univ Paris 07, Unité de recherche clinique, UMR 942, Paris, France

* adrien.six@sorbonne-universite.fr

## Abstract

### Background

Identifying differentially expressed genes between experimental conditions is still the gold-standard approach to interpret transcriptomic profiles. Alternative approaches based on diversity measures have been proposed to complement the interpretation of such datasets but are only used marginally.

### Methods

Here, we reinvestigated diversity measures, which are commonly used in ecology, to characterize mice pregnancy microenvironments based on a public transcriptome dataset. Mainly, we evaluated the Tsallis entropy function to explore the potential of a collection of diversity measures for capturing relevant molecular event information.

### Results

We demonstrate that the Tsallis entropy function provides additional information compared to the traditional diversity indices, such as the Shannon and Simpson indices. Depending on the relative importance given to the most abundant transcripts based on the Tsallis entropy function parameter, our approach allows appreciating the impact of biological stimulus on the inter-individual variability of groups of samples. Moreover, we propose a strategy for reducing the complexity of transcriptome datasets using a maximation of the beta diversity.

### Conclusions

We highlight that a diversity-based analysis is suitable for capturing complex molecular events occurring during physiological events. Therefore, we recommend their use through the Tsallis entropy function to analyze transcriptomics data in addition to differential expression analyses.

**Data Availability Statement:** The microarray dataset used in this paper was initially published and analyzed in Nehar-Belaid et al. (Nehar-Belaid et al. 2016) and is available on Gene Expression

Omnibus repository under accession number GSE68433.

**Funding:** This work has been funded by Assistance Publique-Hôpitaux de Paris, Sorbonne University, Inserm, LabEx Transimmunom (ANR-11-IDEX-0004-02).

**Competing interests:** The authors do not declare any conflict of interest.

## Introduction

Transcriptomic analyses are mainly based on the interpretation of differentially expressed genes (DEG) [1–3]. However, complementary methods have been developed to capture and interpret the complexity of transcriptome datasets. Gene-set analysis (GSA) is one of them and is particularly important in the field of systems biology to describe the regulation of groups of genes involved in a function of interest [4–6]. The abstraction level of analyses can be increased even more with the use of information theory-based entropies and their derived diversity indices. Indeed, diversity indices are widely used measurements describing complex systems such as ecological systems and human microbial communities [7, 8].

Analyzing the diversity of a transcriptomic profile is similar to looking at the quantity of information it contains. This diversity reaches its maximum when all transcripts have the same abundance and reaches its minimum when only one transcript is measured. Diversity-based analyses provide high-level information about transcript abundances within a given biological sample and its similarities with other samples. However, the use of diversity indices remains confidential for transcriptome analysis and restrained to few well-known indices (*e.g.*, Richness, Shannon, or Simpson).

The originality of the work that we present here is the analysis of heterogeneous multicellular transcriptome profiles using the Tsallis entropy function [9], which is a generalized entropy function of order q, equivalent to the Rényi function [10].

Here, we hypothesized that varying the order of the Tsallis entropy function allows for capturing the finest structures in transcriptome datasets and reveals information on the inter-individual variability. Therefore, we used the Tsallis entropy function on a transcriptome dataset initially analyzed using traditional approaches (DEG and GSA) [11]. Our approach enriched the interpretation of this dataset both at the whole transcriptome and gene signature levels.

We demonstrate that the use of the Tsallis entropy function provides additional insights on the transcriptome behaviour compared to traditional approaches or approaches based on the Shannon entropy solely. In turn, we propose this diversity-based analysis to extract the most informative transcript subsets from whole transcriptome datasets as a manner to improve the interpretation of transcriptome datasets.

## Material and methods

### Transcriptomic dataset used in this study

The transcriptomic dataset used in this study was initially published and analyzed in Nehar-Belaid et al. (Nehar-Belaid et al. 2016). This dataset contains transcript abundances of entire uteri from non-pregnant mice (NP), and groups of 3–4 mice at days 4, 6, 8, 10, 11, and 12 of pregnancy. Raw data are available on the Gene Expression Omnibus repository under accession number GSE68433.

### Transcriptomic profiling and data processing

As detailed in Nehar-Belaid et al. [11], RNA from the entire uteri was extracted using a tissue lyser (Qiagen) and TRIzol (Invitrogen) protocols. RNA integrity was assessed using a Bioanalyzer (Agilent). RNA was amplified using the Illumina TotalPrep RNA kit, then analyzed on the MouseWG-6 v2.0 microarrays (Illumina). Raw probe expression measurements were extracted using BeadStudio software (Illumina). Probes with a detection p-value above 0.001 in all transcriptomic profiles were removed from the study. Gene expression data were normalized using the quantile method. Normalized transcript expression was then

log2-transformed and analyzed for differential expression using modified t-test from *limma* package. The final transcript expression matrix comprised 12,375 transcripts across all profiles.

Potential measure biases due to batch effect or other technical steps were evaluated using the Principal Variation Component Analysis method from the package *sva*, and no known bias was found (**S1A Fig**). PVCA analysis was reapplied to normalized data showing an improved profile compared to non-normalized data (**S1B Fig**).

## Extraction of signature-derived down- and up-regulated gene lists

In this paper, in addition to reinvestigating the whole transcriptome datasets during pregnancy in Nehar-Belaid et al. [11], we focused on genes belonging to identified down- or up-regulated signatures at days 6 and 12. To that end, as schematized in **S2 Fig**, we collected the 31 down- and 33 up-regulated signatures on day 6 (depicted in Fig 2 in Nehar-Belaid et al.). Down-regulated signatures are described as enriched in immune-related genes, whereas the up-regulated signatures are enriched in cell-cycle- and proliferation-related genes. We compiled the lists of genes belonging to these signatures to create the union of down- and up-regulated signature genes, after filtering-out genes belonging to both groups, based on the sign of the *limma*'s modified t-test score. The remaining genes were assigned to a control group, named *OTHER*. We thus obtained 1,538 genes in the down-regulated gene list day 6 *DN* genes, 1,425 in the up-regulated gene list day 6 *UP* genes, and 9,281 in the control group list day 6 *OTHER*. We applied the same approach for the 81 down- and 42 up-regulated day 12 signatures depicted in Nehar-Belaid et al. and generated the 685-gene day 12 *DN*, 752-gene day 12 *UP* and 10,938-gene day 12 *OTHER* lists.

## Diversity analysis

Diversity analyses were performed on R using the *entropart* package and its function *DivEst* with default settings, except for the number of simulations that was set to 100 to provide confidence interval of the estimations [12]. The function *DivEst* takes as input a metacommunity object, e.g. a group of transcriptomic profiles belonging to the same kinetic point, derived from the non-transformed transcript expression values. It estimates the alpha and gamma diversities of a meta-community as an effective number of transcripts, and beta diversity as an effective number of profiles.

**Gene Set Enrichment Analysis.** First, the *UP*, *DN* and *OTHER* lists were tested for functional enrichment by GSEA against the overall transcriptome dataset based on their statistical difference as compared to controls. **S3 Fig** shows how the different subsets at day 6 and 12 are effectively enriched as expected: the *DN* subset is biased for down-regulated genes, whereas the *UP* subset is biased towards up-regulated genes; the *OTHER* subset shows no significant enrichment.

Gene signature analysis was performed using Gene Set Enrichment Analysis (GSEA) [13]. GSEA randomizations were performed 1,000 times at the signature level.

**Use of the beta diversity to decomplexify transcriptomic datasets.** We used the following approach, based on the beta diversity, to extract informative part of transcriptomic datasets:

1. The *DivEST* function was applied to calculate $\beta_q$, the beta diversity for the meta-community including all transcriptomic profiles, with increasing values of parameter $q$ between 0 and 5 by steps of 0.01.

2. From the list of $\beta_q$ obtained, a $q_{max}$ parameter was then determined as the $q$ value maximizing $\beta_q$ for the dataset.

3. For $\alpha_{qmax}$, the *DivEST* function provided $\alpha_{qmax}$, the alpha diversity of each group of profiles. Each $\alpha_{qmax}$ represented the effective number of transcripts $n$ yielding the observed diversity. We then used this number to retain the $n$ most abundant transcripts for each group of profiles.

4. The merged gene list comprised the union of the selected transcripts for each group of profiles resulting in a final list of $m$ unique transcripts.

5. A filtered dataset was then created from the original dataset matching this $m$ transcript list.

6. Comparative Principal Component Analysis was finally performed on the original dataset and the filtered dataset.

## Results and discussion

### Rationale and analytical strategy

To test our hypothesis that varying the order of the Tsallis entropy function allows capturing relevant information in transcriptomes, we used a dataset from a study published by our laboratory focusing on transcript expression patterns during pregnancy in uteri of C57BL6 mice [11]. The dataset of Nehar-Belaid et al. contains transcript abundances of entire uteri from non-pregnant mice (NP), and groups of 3–4 mice at days 4, 6, 8, 10, 11, and 12 of pregnancy.

In Nehar-Belaid et al., we demonstrated that the gestation process induced strong differences in terms of transcript abundance between all time-points, as demonstrated by PCA representation in **Fig 1**. Samples from the different time-points are largely non-overlapping to each other, highlighting the specificity of each sample group. In the same study, a gene set-based approach identified molecular signatures further characterizing the biological processes occurring during the gestation, with the evolvement of different immune cell lines. This observation gave credit to the use of gene set-based analyses in situations where high numbers of genes are significantly modulated (more than 5,000 genes when comparing later time points, E8-E12, to non-pregnant mice group, NP).

As looking at molecular signatures provides complementary information to individual gene analysis, we seek to obtain a measure capturing information from an even higher scale than molecular signatures, namely the whole transcriptome itself. In this paper, we took this approach one step further by looking at the whole transcriptome level using diversity indexes.

This entropy-based approach was previously used to analyze transcript abundance profiles of whole organs [14] and was relevant for transcriptome analysis.

In this work, the authors categorized organs based on the diversity of their transcript abundance. They also defined specificity and specialization scores characterizing the different organs.

Two aspects make our work complementary to the study of Martinez and Reyes-Valdes. First, the expression variations that we are studying are not originating from different tissues having specific biological roles but from a unique and complex tissue at different stages of a biological process. In other terms, we are looking at the effects of biological stimulation within a single tissue. Second, Martinez and Reyes-Valdes focused on Shannon's entropy to describe the overall expression of genes in the tissue. In our study, we are extending the analysis to

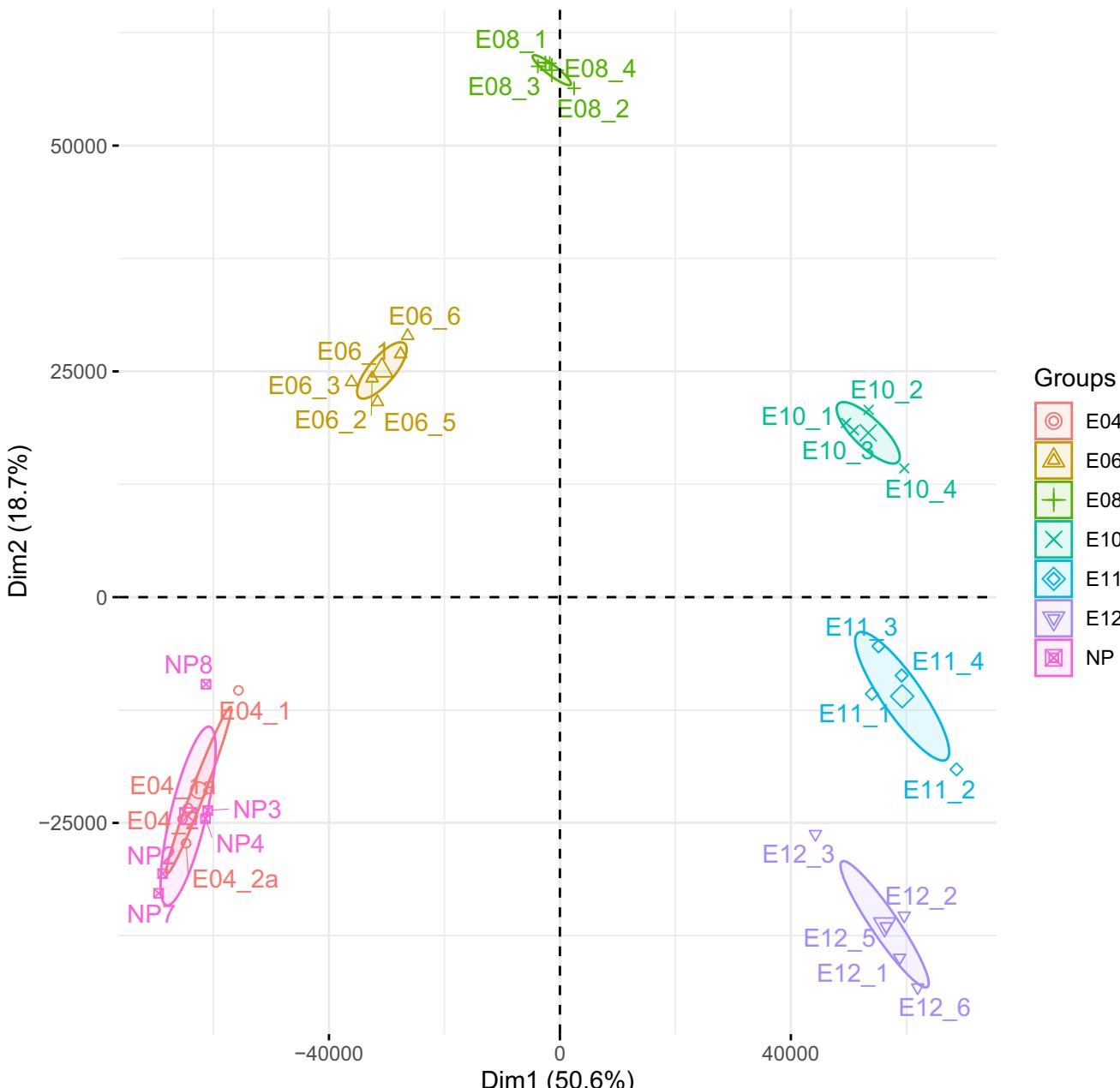

**Fig 1. Principal component analysis.** Transcriptome analyses were performed on non-pregnant mice (NP), and at days 4, 6, 8, 10, 11, and 12 *post coitum* as reported in Nehar-Belaid et al. 2016. Principal component analysis was performed on the original normalized dataset comprising 12,475 genes.

other entropy measures by using the Tsallis entropy, a function of order q. Varying the parameter q allows for looking at different entropy measures. Then we transformed these entropies into alpha diversity measures to ease biological interpretations as the measures now refer to an effective number of genes. Noteworthy, we focus in this work on the analysis of sample groups rather than individuals. We also introduce the use of the measure of the beta diversity, expressed as an effective number of transcriptomic profiles, as evaluating the variability of transcriptomes between individuals of the same group.

## Theoretical framework

We propose to use the Tsallis entropy function, described in **Eq 1**, and its transformation, described in **Eq 2,** to evaluate the alpha diversity of a transcriptomic profile.

$$_j^q H_\alpha \equiv \frac{1 - \sum_{i=1}^S p_{ij}^q}{q - 1},$$

Eq 1

where $p_{ij}$ is the relative frequency of a transcript $i$ in a given profile $j$, $S$ is the number of different transcripts within this profile, and $q$ is the order of the equation.

Particular values of $q$ in **Eq 1** correspond to usual diversity measures. When $q$ is equal to 0, then $_j^q H_\alpha$ corresponds to the richness of the transcriptome. When $q$ is equal to 1 or 2 then $_j^q H_\alpha$ respectively corresponds to the Shannon's or Simpson's entropy. When $q$ tends to infinity, then $_j^q H_\alpha$ corresponds to the Berger-Parker index, the relative expression of the most abundant transcript in the profile.

Due to the fact that $_j^q H_\alpha$ is unitless, entropy measures can be difficult to interpret. Thus, Jost and colleagues [15] proposed to transform these measures into an effective number of species (transcripts in our case) by using the exponential of the entropy as described in **Eq 2**.

$$_j^q D_\alpha = e_q^{_j^q H_\alpha}$$

Eq 2

Complementary to the alpha diversity, the beta diversity can be used to quantify the diversity between profiles of the same group. A beta diversity of 1 indicates that there is no composition difference between profiles in terms of transcript abundance. A beta diversity of 2 indicates that the overall composition difference of the sample groups is equal to that of two samples with no common transcripts.

The beta diversity of a set of profiles is given by the **Eq 3,** which correspond to the ratio of the gamma diversity of the set of profiles to its alpha diversity [16].

$$^q D_\beta = {}^q D_\gamma / {}^q D_\alpha$$

Eq 3

In this study, we investigated the alpha and beta diversity distributions of transcriptomic profiles through the kinetic of gestation in a C57BL6 model based on different orders of diversity.

## Exploration of the transcriptome alpha diversity

We analyzed alpha diversity measures, as a function of the $q$ parameter as defined in **Eq 2** in a transcriptomic dataset of uterine microenvironment during pregnancy at the different time-points of the kinetic (days 4, 6, 8, 10, 11 and 12 *post coitum*) and for a control group of non-pregnant mice (NP).

Increasing the order of the Tsallis entropy function gives more weight to highly expressed genes, gradually ignoring low-expressed genes in the calculation of the alpha diversity. When $q = 0$, the alpha diversity is equal to the number of genes within a given set of profiles (and corresponds to the richness), which is the same in all profiles in the dataset (12,375).

**Fig 2** shows the evolution of the alpha diversity for each time-point using different values of $q$ ($q \in \{0.5,1,2,3,4,5\}$). When $q$ equals 0.5 or 1 (Shannon's index), the alpha diversity is quite similar between NP, day 4, and day 6. The alpha diversity then drops significantly at day 8, and rebounds at day 10 until day 12 (**Fig 2A and 2B**). For $q$ equals 2, 3, and 4, this pattern is different for the earlier time-points, as the group day 4 unhooks the conditions NP and day 6 (**Fig 2C–2F**). Finally, when $q$ is equal to 5, the distributions of alpha diversity values overlap at day 4 and day 8 (**Fig 2F**). Overall, the behaviour of the alpha diversity after day 8 remains constant, with a rebound at day 10 and day 11 and a decrease at day 12 observable when $q > 0.5$.

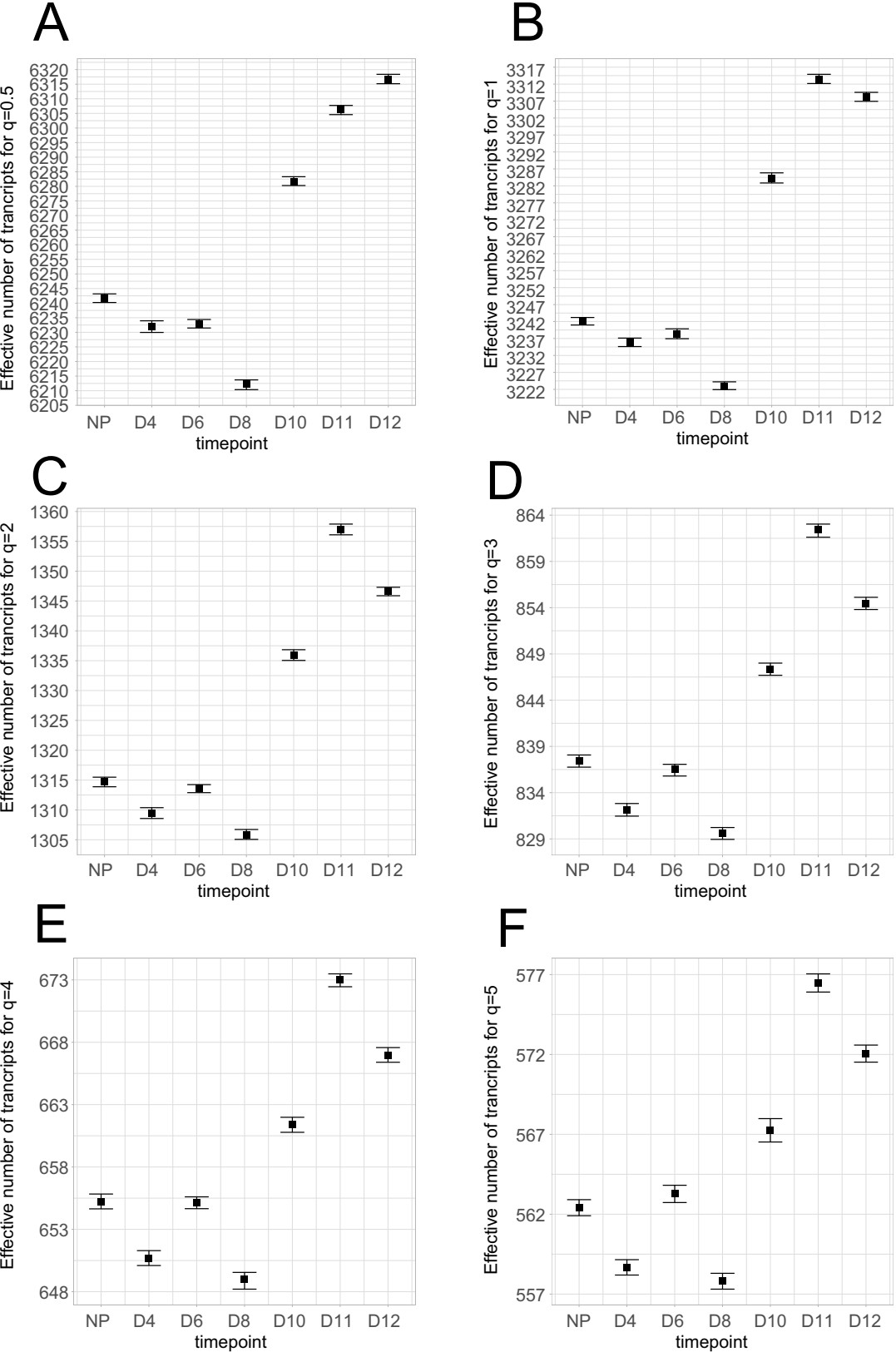

**Fig 2. Alpha diversity analysis as a function of *q*.** Transcriptome analyses were performed on non-pregnant mice (NP), and at day 4 (D4), day 6 (D6), day 8 (D8), day 10 (D10), day 11 (D11), and day 12 (D12) as reported in (Nehar-Belaid et al. 2016). For each set of profiles, alpha diversity is calculated for different values of the Tsallis function *q* parameter: (**A**) $q = 0.5$, (**B**) $q = 1$ (Shannon), (**C**) $q = 2$ (Simpson), (**D**) $q = 3$, (**E**) $q = 4$ and (**F**) $q = 5$. Bars represent the 95% bootstrap confidence interval calculated from 100 iterations where profiles are simulated from a multinomial distribution following the observed transcript frequencies.

Tohmeresz states that a diversity can be considered as higher than another when this alpha diversity is higher than the other for all possible values of *q* [17]. In our experiment, the conditions NP and day 6 cannot be ordered as the relative diversity rank changes after $q = 3$. On the contrary, the rebound after day 8 is consistent for the different values of *q*. Therefore, we can state that the alpha diversity is increasing after day 8 in our model. Interestingly, the alpha diversity at day 12 is systematically below that of day 11 when $q > 0.5$.

Altogether, these results highlight the importance of looking at different values of the parameter *q* when interpreting alpha diversity by providing more detailed information about the structure of highly expressed genes. A biological interpretation of these results is that there is a consistent behaviour of the uterine transcriptome during pregnancy. On day 8, we observed a contraction of the alpha diversity, which means that fewer highly expressed genes are taking more importance. Then, the alpha diversity increases until day 11, highlighting that new genes are highly expressed.

## Exploration of the transcriptome beta diversity

We assessed the beta diversity within our dataset as a function of the *q* value (*q* ranging from 0 to 5) of the Tsallis entropy function using the same transcriptomic dataset (**Fig 3**).

For each time-point, we calculated the beta diversity capturing the variability of the composition of the profiles. Indeed, one can see the beta diversity as the interindividual variability within each group. The beta diversity measure provides a single value starting from 1, when profiles share the same composition, to a maximum equal to the number of profiles when profiles are drastically not sharing the same composition.

We observed that the beta diversity of the NP condition increased until $q = 1.5$ followed by a clear decrease. The beta diversity curves for the set of profiles at days 4, 6, and 8 tend to stabilize their beta diversity for $q > 1.5$. Conversely, beta diversity values at days 10, 11, and 12 increase with *q*. The decrease of NP beta diversity when *q* is high can be explained by the nature of the biological samples analyzed. A biological interpretation of these results is that the uterus is not as specialized as other organs [14], meaning that it expressed a broad variety of transcripts. The most abundant transcripts captured when $q = 5$ tend to be the same. In addition, these genes are expressed more evenly across individuals, compared to what is found for an intermediate diversity index (*e.g.*, $q = 2$).

The relative positions of beta diversity curves drop gradually, from NP until day 8, and then rise from day 10 until day 12. These results suggest that early pregnancy groups tend to homogenize their gene expression. This also results in the modulation of a group of shared genes, therefore decreasing the beta diversity. An interpretation of these observations is that the activation of a group of genes is preparing the potential fetus implantation.

After day 8, the beta diversity rises, a behaviour that can be explained by the increasing importance of genes needed for the foetus development. The production of a vast panel of cell types leads to an increase of the alpha diversity but also of the beta diversity, in relation to slight fluctuations of fetus development between individuals. Interestingly, this statement holds true for all *q* values, which means that this behaviour is not dependent on the gene expression levels.

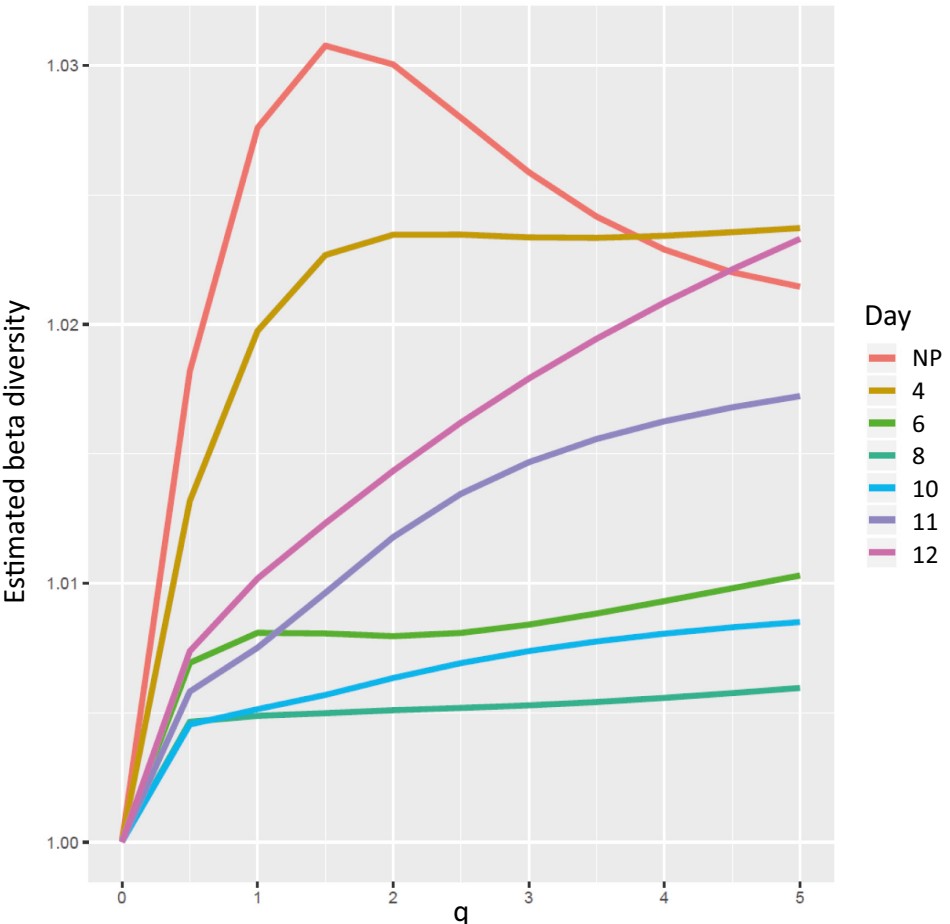

**Fig 3. Beta diversity analysis as a function of $q$.** Transcriptome analyses were performed on non-pregnant mice (NP), and at day 4 (D4), day 6 (D6), day 8 (D8), day 10 (D10), day 11 (D11), and day 12 (D12) as reported in Nehar-Belaid et al. 2016. For each set of profiles, beta diversity is calculated with *DivEst* function for $q$ values between 0 and 5 with a step of 0.5. A beta diversity value of 1 corresponds to profiles having the same diversity for the given time-point.

As a conclusion from these results, we found that the alpha and beta diversity analyses are complementary in the interpretation of this transcriptomic dataset. Indeed, the alpha diversity depicts how the uterus, dedicated to a specific and critical biological function, is impacted by stimulation. Our results suggest that it is moving from a relaxed state (NP) to a biologically constrained state (day 8) and increasing again due to fetal development. The beta diversity confirms this observation, revealed as early as day 4. Moreover, our analysis shows that fetal development increases the beta diversity, particularly when looking at higher expressed genes ($q \geq 1.5$).

## Exploration of diversity indices within differentially expressed gene signatures

Considering these results obtained on the whole transcriptome datasets, we applied the same methodology restricted to the list of genes derived from up- or down- regulated signatures between day 6 or day 12 samples relative to the non-pregnant condition.

For each of these two sub-datasets, three different gene subsets were created: (i) the up-regulated genes (*UP*) as compared to NP; (ii) the down-regulated genes as compared to NP (*DN*);

and (iii) non differentially expressed genes (*OTHER*). The gene subsets were derived from the results obtained by Nehar-Belaid et al. [11]. The 31 down-regulated molecular signatures were used to define the collection of down-regulated genes (*DN*) while the 33 up-regulated molecular signature were used for the up-regulated genes list (*UP*). Other contains the remaining genes (see **S2 Fig** and Material and Method section). Then, we compared the Shannon and Simpson alpha diversity indices obtained for day 6 versus NP, and for day 12 versus NP.

As shown in **S4 Fig**, the *DN* and *UP* subsets showed a similar behaviour for the day 6 vs. NP comparison. This observation is consistent with the overall analysis depicting a decrease of diversity in day 6 sample group compared to NP sample group. Even if the absolute and relative changes are modest, they remain meaningful. Noteworthy, the *OTHER* subset shows a slight increase at day 6 relative to NP, considering that it summarizes the behaviour of 10-fold more genes than *UP* and *DN* subsets. Again, the observed decrease of alpha diversity at day 6 can be explained by the constraints applied on the uterus when fertilization and implantation of the fetus occur (see section Exploration of the alpha transcriptome diversity indices).

In contrast, the day 12/NP *DN* and *OTHER* subsets see their alpha diversity increasing at day 12 compared to NP, in line with the previously observed overall behaviour, when the *UP* subset decreases at day 12 (**S5 Fig**). This latter observation is remarkable since the *UP* subset behaviour was hidden by that of *DN* and *OTHER* subsets in the global analysis.

Altogether, these analyses suggest that at day 6, both biological processes that comprise up- (cell-cycle & proliferation) and down- (immunological processes) regulated genes follow the same trend for diversity decrease related to genes associated with these biological processes being activated. On day 12, the overall diversity increase can be explained by an increase in cell population diversity due to the fetal development and the migration of immunological cells, including regulatory T-cells, immature dendritic cells, myeloid-derived suppressor cells (MDSCs), NK or NKT cells [18–22]. On the contrary, the *UP* subset behaviour remains stable between days 6 and 12, which can be explained by the cell cycle and proliferation mechanisms staying at work at the level of the organism.

We then looked at the comparative beta diversity analysis. The beta diversity remains very low for all subsets in the day 6/NP comparison (**Fig 4A**). This means that individual transcriptomic profiles are not diverging much from each other and are similar in terms of transcript composition. Interestingly, for the day 12/NP comparison (**Fig 4B**), the beta diversity of *DN* subset shows a strong increase at day 12, when that of *UP* samples shows a modest increase rapidly reaching a plateau. This can be explained when considering our previous observation related to the strong increase of alpha diversity at day 12 for *DN* subset: along with fetal development, various immune cell populations are recruited to ensure a proper uterine tolerogenic environment in an asynchronous manner explaining why individual samples tend to diverge from each other.

Altogether, we conclude here that beta diversity analysis of transcriptome datasets is interesting as it provides novel insights on the behaviour of samples or groups of samples considering different levels of transcript abundance. The estimation of alpha and beta diversities for different values of *q* shown patterns that differ when giving more or less influence on the most abundant transcripts. This implies changes in terms of transcripts composition and number, and inter-individual variability. These changes can be explained biologically and therefore enrich the analysis of such datasets.

## On the use of beta diversity to decomplexify transcriptome datasets

Based on these last results, we assessed whether the estimation of the beta diversity could be used as a statistical, rather than heuristic, means to reduce the datasets to a minimum number

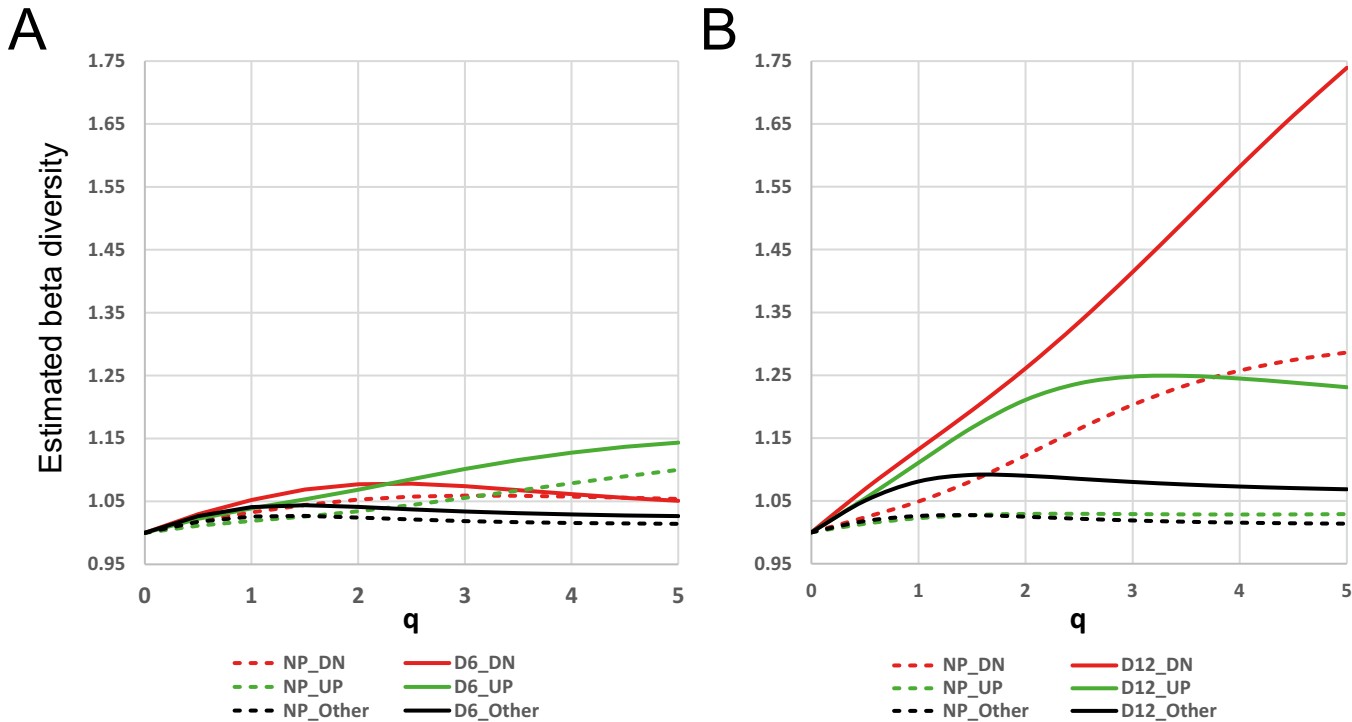

**Fig 4. Beta diversity analysis.** For each UP (red), DN (green), and Other (black) subsets (see **S2 Fig** and Material and Methods section), the beta diversity was calculated across profiles belonging to NP (dashed line), and D6 (**A**) or D12 (**B**) samples (plain line), respectively, with the *DivEst* function as a function of $q$ comprised between 0 and 5 with a step of 0.5. A value of 1 means that profiles at this time-point are similar in terms of diversity, thus equivalent to their average.

of transcripts, while conserving a maximum of information. Therefore, we looked at finding the maximum of the beta diversity within the dataset while ranging different values of $q$.

Indeed, beta diversity describes how much profiles in a set of profiles are different in terms of transcript abundance. Hence, this approach can be used to filter out transcripts that participate less to the diversity of dataset and considered as less informative. Therefore, we searched for the value of $q$ that maximizes the beta diversity for the whole dataset and used this $q_{max}$ to calculate the effective number of transcripts that should be retained in each set of profiles.

To do so, we first performed a beta diversity analysis to determine the $q_{max}$ value reaching the maximum beta diversity across the entire dataset. The beta diversity reached a maximum of 1.08 when $q = 1.88$, thus setting $q_{max}$ to 1.88. Of note, this value lies between the Shannon ($q = 1$) and Simpson ($q = 2$) diversity indices. Then, as detailed in the Material & Methods section, the alpha diversity $q_{max}$ of each set of profiles at $q_{max}$ is estimated providing the $m$ transcripts to retain for each set of profiles. Transcripts are ordered based on their expression level and the first $m$ most-expressed transcripts are selected. Finally, the retained genes from the different set of profiles were merged into a single list of 2,161 transcripts.

To test the relevance of this approach, we compared the projections obtained by two principal component analyses (PCA) applied both on the original transcriptomics dataset and on the dataset restricted to the 2,161 selected transcripts. As shown in **Fig 5**, the positioning of sample groups on the first two components is very consistent between the two analyses. This indicates that the information discriminating sample groups is very well captured using this selected gene list, while being six times smaller than the original dataset.

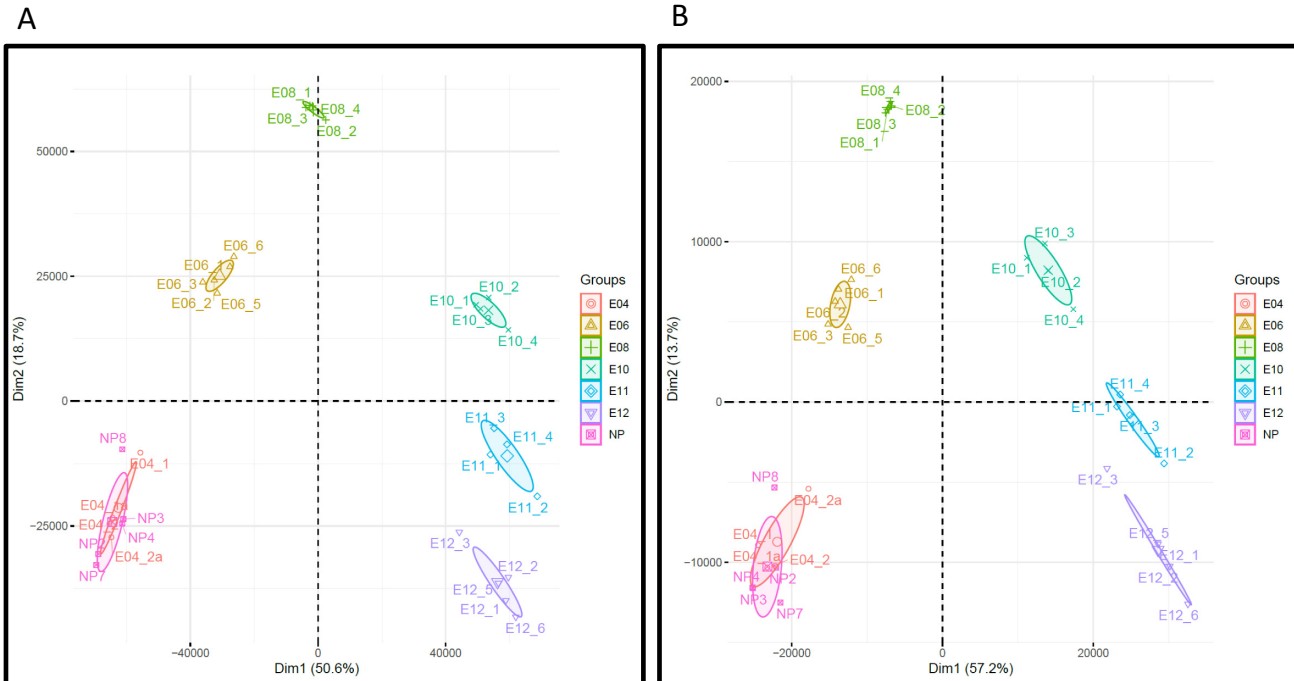

**Fig 5. Impact of transcript filtration based on the beta diversity measure.** Transcriptome analyses were performed on non-pregnant mice (NP), and at days 4, 6, 8, 10, 11, and 12 *post coitum* as reported in Nehar-Belaid et al. 2016. Principal component analysis was performed on the original dataset comprising 12,475 transcripts (**A**), and on the decomplexified dataset comprising the 2,161 transcripts obtained based on beta diversity maximization method described (**B**).

Our results are similar to those obtained by Ogata et al. in a very different setup [23]. In their study, the authors evaluated the relationship between environmental changes and the amount of transcriptome change in silkworm fat-body tissues cultivated with phenobarbital. The authors showed that the transcriptome diversity, evaluated by the Shannon entropy, decreases under drug treatment and that the 500 genes with the highest relative frequency well summarized the overall transcriptome behaviour. However, their approach for determining this number of essential genes was heuristic, while the advantage of the approach that we propose relies on a systematic maximization of a mathematically defined diversity index.

## Conclusions

Recently, with the advent of single-cell RNA analysis, diversity, including Tsallis entropy function, and similarity indices were implemented for characterizing the heterogeneity of tumour cells, implying potential consequences on treatment efficacy [24, 25].

Information theory-based method has also been evaluated to decipher high-level structures of transcript abundance profiles of whole organs [14]. In this study, the authors used the Shannon entropy to characterize the diversity of transcriptomes coming from 28 different human tissues. The Shannon entropy was also implemented to study gene variation detection in multiple samples and shown its complementarity with more traditional approaches [26]. In the last two cases, diversity was measured using the Shannon entropy, a well-known equation in scientific fields like physic or informatics to provide a measure of the quantity of information in a system.

In this work, we generalized the application of information theory diversity indices initially described by Martinez and Reyes-Valdes [14] to investigate the alpha and beta diversities of murine pregnancy transcriptome datasets that we previously produced and analyzed [11] in function of a parameter $q$.

This previous work, done at the level of genes or gene signatures, explored the transcriptional changes during fetal development at different time-points *post coitum* (days 4, 6, 8, 10, 11, and 12) as compared to non-pregnant mice (NP). This study assessed how the immunoregulatory balance between regulatory and effector T cells is shaped and revealed striking similarities taking place at the very first days after tumour or embryo implantation, suggesting that the mechanisms that protect mammalian fetuses from immune attack are diverted during tumor development. Indeed, in both situations, up-regulated signatures are mostly linked to DNA and RNA synthesis, ribosome, proteasome, and cell cycle when down-regulated signatures are related to immune system processes [11].

Our generalization relies on the use of the Tsallis entropy function functions that allow estimating diversity indices with relative consideration of low- vs. high-expression genes [10]. We described the behaviour of the overall transcriptome data showing an initial contraction of the alpha diversity at day 8 followed by an increase as compared to non-pregnant mice. This phenomenon can be explained by the contraction of gene expression toward a common set of genes related to pregnancy in the early phase; in the later phase, high-expression genes tend to be less synchronously expressed across samples.

We then focused our analysis on the genes belonging to immune system-related downregulated or cell cycle-related up-regulated gene signatures at days 6 and 12 of pregnancy as demonstrated in our initial study [11]. Interestingly, on day 12, we reveal a different behaviour of the up-regulated genes, the diversity of which decreases as opposed to the overall transcriptome change related to fetus development, when both up- and down- regulated gene diversity values decrease at day 6. We thus demonstrate the advantage of combining a global diversity index approach with gene signature selection to reveal transcript subset differential behaviour related to a biological process.

Finally, we propose an approach relying on the maximization of beta diversity to extract the most informative transcript subsets from whole transcriptome datasets to improve transcriptome analysis. Although ignoring up to 85% of the original information can be subject to debate, this approach certainly offers a means of reducing the necessary number of transcripts to consider, thus reducing computing time and power for machine-learning approaches, for example. It should also be effective to reduce the complexity of datasets before unsupervised analysis like independent component analysis (ICA) or to serve as a feature selection step in machine learning approaches, reducing the noise produced by consistently low-expressed genes.

## Supporting information

**S1 Fig. Data quality by PVCA before and after normalization.** Dataset is tested for the presence of technical biases using principal variation component analysis (PVCA). Seven possible sources of batch effect were evaluated, in addition to the 'kinetic' biological parameter. Residual is the part of the variance not explained by the different parameters tested (**A**). After normalization, 96.13% of the variance is explained by the kinetic, demonstrating the low impact of technical sources on the dataset (**B**).
(PDF)

**S2 Fig. E6 data subset creation.** A: Genes are extracted from day 6 regulated molecular signatures from Nehar-Belaid et al. 2016. B: Genes belonging to both 33 *UP* and 31 *DN* gene lists

are dispatched based on their eBayes score calculated by comparing gene expression at day 6 to Controls (NP). C: The *OTHER* subset is created by removing *UP* and *DN* subset from the full dataset. Similarly, genes from the 81 down- and 42 up-regulated signatures at day 12 were extracted to build the *DN* (685 genes), *UP* (752 genes) and *OTHER* (10,938 genes) gene lists. (PDF)

**S3 Fig. Enrichment analysis of day 6 and 12 compared to NP *DN*, *UP* and *OTHER* subsets.** GSEA analysis provided statistical analysis of gene expression biases in the *DN* (**A, D**), *UP* (**B, E**) and *OTHER* (**C, F**) subsets constructed as described in S2 Fig and Material and Methods section. A, E: *DN* subset shows significant enrichment for down-regulated genes (q-value$<10^{-7}$). B, E: *UP* subset shows significant enrichment for up-regulated genes (q-value$<10^{-7}$). C, F: *OTHER* subset shows no enrichment (q-value$>0.05$). (PDF)

**S4 Fig. Alpha diversity analysis of day 6/NP *DN*, *UP* and *OTHER* subsets.** For each subset, Shannon (**A, C and E**) and Simpson (**B, D and F**) alpha diversity were calculated as the average of alpha diversities of individual profiles for the 6/NP *DN* (**A and B**), *UP* (**C and D**) and *OTHER* (**E and F**) subsets. Bars represent the 95% bootstrap confidence interval calculated from 100 iterations where profiles are simulated from a multinomial distribution following the observed transcript frequencies. (PDF)

**S5 Fig. Alpha diversity analysis of day 12/NP *DN*, *UP* and *OTHER* subsets.** For each subset, Shannon (**A, C and E**) and Simpson (**B, D and F**) alpha diversity were calculated as the average of alpha diversities of individual profiles for the 12/NP *DN* (**A and B**), *UP* (**C and D**) and *OTHER* (**E and F**) subsets. Bars represent the 95% bootstrap confidence interval calculated from 100 iterations where profiles are simulated from a multinomial distribution following the observed transcript frequencies. (PDF)

**S1 Graphical abstract.** (PDF)

## Acknowledgments

We thank Wahiba Chaara, Véronique Thomas-Vaslin, Encarnita Ferrandiz for helpful discussions as well as their critical reading of the manuscript.

## Author Contributions

**Conceptualization:** Nicolas Dérian, Djamel Nehar-Belaid, Nicolas Tchitchek.

**Investigation:** Nicolas Dérian, Hang-Phuong Pham, Djamel Nehar-Belaid, Nicolas Tchitchek, David Klatzmann, Vicaut Eric, Adrien Six.

**Methodology:** Nicolas Dérian, Hang-Phuong Pham, Djamel Nehar-Belaid, Nicolas Tchitchek, David Klatzmann, Vicaut Eric, Adrien Six.

**Supervision:** Adrien Six.

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
