## [Decision Letter · Decision Letter 0]

22 Nov 2021

PONE-D-21-29524The Tsallis generalized entropy enhances the interpretation of transcriptomics datasetsPLOS ONE

Dear Dr. Six,

Thank you for submitting your manuscript to PLOS ONE. After careful consideration, we feel that it has merit but does not fully meet PLOS ONE’s publication criteria as it currently stands. Therefore, we invite you to submit a revised version of the manuscript that addresses the points raised during the review process. According to referee's sugestions (see detailed comments below), the manuscritp should be improved in several aspects. Please, do not forget that data and scripts must be fully available.

We look forward to receiving your revised manuscript.

Kind regards,

Francisco J. Esteban, Ph.D., M.Sc.

Academic Editor

PLOS ONE

Journal Requirements:

"This work has been funded by Assistance Publique-Hôpitaux de Paris, Sorbonne University, Inserm, LabEx Transimmunom (ANR-11-IDEX-0004-02). We thank Wahiba Chaara, Véronique Thomas-Vaslin, Encarnita Ferrandiz for helpful discussions as well as their critical reading of the manuscript"

"This work has been funded by Assistance Publique-Hôpitaux de Paris, Sorbonne University, Inserm, LabEx Transimmunom (ANR-11-IDEX-0004-02)"

3. We note that you have included the phrase “data not shown” in your manuscript. Unfortunately, this does not meet our data sharing requirements. PLOS does not permit references to inaccessible data. We require that authors provide all relevant data within the paper, Supporting Information files, or in an acceptable, public repository. Please add a citation to support this phrase or upload the data that corresponds with these findings to a stable repository (such as Figshare or Dryad) and provide and URLs, DOIs, or accession numbers that may be used to access these data. Or, if the data are not a core part of the research being presented in your study, we ask that you remove the phrase that refers to these data

Reviewers' comments:

Reviewer's Responses to Questions

**Comments to the Author**

1. Is the manuscript technically sound, and do the data support the conclusions?

Reviewer #1: Yes

2. Has the statistical analysis been performed appropriately and rigorously? 

Reviewer #1: Yes

3. Have the authors made all data underlying the findings in their manuscript fully available?

Reviewer #1: Yes

4. Is the manuscript presented in an intelligible fashion and written in standard English?

Reviewer #1: Yes

5. Review Comments to the Author

Reviewer #1: The paper introduces diversity measures for the analysis of transcriptomes as new tools providing additional information to previous approaches, including a limited use of diversity indices.

It is quite interesting but really hard to follow, so I suggest to make it more straightforward.

My concern is that it focuses so much on the proposed tools that essential aspects of the data are left aside and their very meaning is less clear than in a less sophisticated analysis.

Figure 4A seems to be the basic view of the data. It shows the evolution of the composition of the transcriptome which is very clear with non-overlapping groups except between non pregnant mice and day 4. It should be presented first: diversity measurements only complete this analysis that would be the only one kept if only a very short presentation was allowed.

Instead of that, the paper's organization makes it cryptic. Without going to methods after the conclusion, it is not possible to be sure that individuals are different mice. I strongly suggest a standard presentation with methods before results and data exploration (i.e. fig. 4A) before the proposed new analyses.

You must explain clearly what diversity means with respect to the data, not only in general. Here, alpha diversity is that of transcriptomes in an average mouse (strictly spreaking, the average diversity of the transcriptomes of the mice of the same group). Beta diversity describes how different from their assemblage these transcriptomes are, i.e. the variability of transcriptomes between mice of the same group.

I'm not convinced by the presentation of beta diversity results.

First, there is a confusion in the introduction of "Exploration of the beta transcriptome diversity indices". Please go back to Jost(2007) cited in Marcon et al. (2014) that you cite. Beta diversity does not capture the variability of alpha diversities. Actually, there is only one alpha diversity that is the average of the diversity of samples. When you write "alpha diversities", you mean "sample diversities". This is not a serious issue but it can be corrected easily.

In the effective number partitioning, gamma diversity is an effective number of transcripts equal to the product of alpha diversity (another effective number of transcripts) and beta diversity, an effective number of samples. Here the sample diversities are exactly the same by definition, and beta diversity is ideally independent from alpha. Beta diversity captures the variabilty of the composition of samples: how different they are from one another. You write it in line 266.

The confusion arises again line 249: the assemblage gamma diversity is derived from is not the average of samples. Actually, samples diverge from one another (so their assemblage has a higher diversity but this is not the point).

In contrast, the biological interpretation (lines 177-197) is very convincing.

The method you applied to decomplexify the dataset is a main novelty of the paper. It is not detailed in lines 260-270 so it remains cryptic. It confirms my opinion in changing the order of sections, see above.

In the method appendix, it is quite clear. line 414, the meaning of "sum" is not clear. Do you mean "set"?

The interpretation of alpha diversity at q_max is confusing. It is an effective number of genes, i.e. the number of genes equally expressed that would yield the observed diversity. It is not a threshold allowing to detect the most expressed ones: there is probably a very small difference between the D_alpha-th and the D_alpha+1-th gene (ordered from the most to the least expressed). So your choice is reasonable but not obvious. You should make it clearer.

In summary, I recommend major revision. Please rewrite the paper straightforwardly with methods before results, insist less on the novelty of the methods beforehand but be more factual and show that diversity analyses allow to complete previous knowledge (but present it before completing it). You may claim novelty when you interpret the results, showing they were not reachable before.

Minor comments

Please call diversity diversity and entropy entropy. For instance, page 6, paragraph starting line 106: you give the partitioning of entropy in the equation but you write diversity in the first sentence. Then, you write that beta entropy (this is correct) can be transformed into an effective number of samples. This number is beta diversity but the reader is confused by the previous sentence. Line 104, you write that eq.3 describes "beta diversity entropy": this is not correct and very confusing: it describes beta entropy.

You can write "diversity of order q": you don't have to refer to entropy, the order applies to both without any ambiguity. e.g., line 125: Increasing the order of diversity gives more weight to highly expressed genes.

In fig. 1, the way confidence intervals are obtained is not clear.

Equation 4 is useless: the good practise is to compute beta diversity as the ratio of gamma to alpha since it is easier and allows using a variety of estimators of diversity (out of the scope of the paper).

Figure 4: change the labels E4, E6, etc. for E04, E06, etc. so that their are sorted correctly in the legend.

6. PLOS authors have the option to publish the peer review history of their article (what does this mean?). If published, this will include your full peer review and any attached files.

Reviewer #1: **Yes: **Eric Marcon

---

## [Author Response · Author response to Decision Letter 0]

7 Mar 2022

March 3rd, 2022,

To the editorial board of PLOS ONE,

Please find the attached our revised manuscript, entitled The Tsallis generalized entropy enhances the interpretation of transcriptomics datasets, by Nicolas Dérian, Hang-Phuong Pham, Djamel Nehar-Belaid, Nicolas Tchitchek, David Klatzmann, Vicaut Eric, Adrien Six, for consideration in PLOS ONE. 

We would like to thank the editor and the reviewer for their valuable feedbacks on our manuscript. We believe that we have addressed all the reviewer’s concerns and we provided point-by-point answers to the reviewer. The modifications that we made on the manuscript are highlighted in yellow.

Sincerely, 

Adrien Six, PhD

---

## [Decision Letter · Decision Letter 1]

24 Mar 2022

The Tsallis generalized entropy enhances the interpretation of transcriptomics datasets

PONE-D-21-29524R1

Dear Dr. Six,

We’re pleased to inform you that your manuscript has been judged scientifically suitable for publication and will be formally accepted for publication once it meets all outstanding technical requirements.

Kind regards,

Francisco J. Esteban, Ph.D., M.Sc.

Academic Editor

PLOS ONE

Additional Editor Comments (optional):

Reviewer: "The authors perfectly addressed my comments on the first version of the manuscript. It is now clear and straightforward."

A correction is needed line 208 : q tends "to infinity" rather than "toward infinite".

Reviewers' comments:

Reviewer's Responses to Questions

**Comments to the Author**

1. If the authors have adequately addressed your comments raised in a previous round of review and you feel that this manuscript is now acceptable for publication, you may indicate that here to bypass the “Comments to the Author” section, enter your conflict of interest statement in the “Confidential to Editor” section, and submit your "Accept" recommendation.

Reviewer #1: All comments have been addressed

2. Is the manuscript technically sound, and do the data support the conclusions?

Reviewer #1: Yes

3. Has the statistical analysis been performed appropriately and rigorously? 

Reviewer #1: Yes

4. Have the authors made all data underlying the findings in their manuscript fully available?

Reviewer #1: Yes

5. Is the manuscript presented in an intelligible fashion and written in standard English?

Reviewer #1: Yes

6. Review Comments to the Author

Reviewer #1: The authors perfectly addressed my comments on the first version of the manuscript. It is now clear and straightforward.

A correction is needed line 208 : q tends "to infinity" rather than "toward infinite".

7. PLOS authors have the option to publish the peer review history of their article (what does this mean?). If published, this will include your full peer review and any attached files.

Reviewer #1: **Yes: **Eric Marcon

---

## [Editor Report · Acceptance letter]

12 Apr 2022

PONE-D-21-29524R1 

The Tsallis generalized entropy enhances the interpretation of transcriptomics datasets 

Dear Dr. Six:

I'm pleased to inform you that your manuscript has been deemed suitable for publication in PLOS ONE. Congratulations! Your manuscript is now with our production department. 

Kind regards, 

on behalf of

Dr. Francisco J. Esteban 

Academic Editor

PLOS ONE